# E-Health and M-Health in Obesity Management: A Systematic Review and Meta-Analysis of RCTs [note 1]

**DOI:** 10.3390/nu17132200

**Published:** 2025-07-01

**Authors:** Manuela Chiavarini, Irene Giacchetta, Patrizia Rosignoli, Roberto Fabiani

**Affiliations:** 1Department of Health Sciences, University of Florence, Viale GB Morgagni 48, 50134 Florence, Italy; manuela.chiavarini@unifi.it; 2Local Health Unit of Bologna, Department of Hospital Network, Hospital Management of Maggiore and Bellaria, 40124 Bologna, Italy; 3Department of Chemistry, Biology and Biotechnology, University of Perugia, 06123 Perugia, Italy; patrizia.rosignoli@unipg.it (P.R.); roberto.fabiani@unipg.it (R.F.)

**Keywords:** e-health, m-health, management, obesity, systematic review

## Abstract

Background: Obesity in adults is a growing health concern. The principal interventions used in obesity management are lifestyle-change interventions such as diet, exercise, and behavioral therapy. Although they are effective, current treatment options have not succeeded in halting the global rise in the prevalence of obesity or achieving sustained long-term weight maintenance at the population level. E-health and m-health are both integral components of digital health that focus on the use of technology to improve healthcare delivery and outcomes. The use of eHealth/mHealth might improve the management of some of these treatments. Several digital health interventions to manage obesity are currently in clinical trials. Objective: The aim of our systematic review is to evaluate whether digital health interventions (e-Health and m-Health) have effects on changes in anthropometric measures, such as weight, BMI, and waist circumference and behaviors such as energy intake, eating behaviors, and physical activity. Methods: A search was conducted for randomized controlled trials (RCTs) conducted through 4 October 2024 through three databases (Medline, Web of Science, and Scopus). Studies were included if they evaluated digital health interventions (e-Health and m-Health) compared to control groups in overweight or obese adults (BMI ≥ 25 kg/m^2^) and reported anthropometric or lifestyle behavioral outcomes. Study quality was assessed using the Cochrane Risk of Bias Tool (RoB 2). Meta-analyses were performed using random-effects or fixed-effects models as appropriate, with statistical significance set at *p* < 0.05. Results: Twenty-two RCTs involving diverse populations (obese adults, overweight individuals, postpartum women, patients with eating disorders) were included. Digital interventions included biofeedback devices, smartphone apps, e-coaching systems, web-based interventions, and mixed approaches. Only waist circumference showed a statistically significant reduction (WMD = −1.77 cm; 95% CI: −3.10 to −0.44; *p* = 0.009). No significant effects were observed for BMI (WMD = −0.43 kg/m^2^; *p* = 0.247), body weight (WMD = 0.42 kg; *p* = 0.341), or lifestyle behaviors, including physical activity (SMD = −0.01; *p* = 0.939) and eating behavior (SMD = −0.13; *p* = 0.341). Body-fat percentage showed a borderline-significant trend toward reduction (WMD = −0.79%; *p* = 0.068). High heterogeneity was observed across most outcomes (*I*^2^ > 80%), indicating substantial variability between studies. Quality assessment revealed predominant judgments of “Some Concerns” and “High Risk” across the evaluated domains. Conclusions: Digital health interventions produce modest but significant benefits on waist circumference in overweight and obese adults, without significant effects on other anthropometric or behavioral parameters. The high heterogeneity observed underscores the need for more personalized approaches and future research focused on identifying the most effective components of digital interventions. Digital health interventions should be positioned as valuable adjuncts to, rather than replacements for, established obesity treatments. Their integration within comprehensive care models may enhance traditional interventions through continuous monitoring, real-time feedback, and improved accessibility, but interventions with proven efficacy such as behavioral counseling and clinical oversight should be maintained.

## 1. Introduction

Obesity in adults is a growing health concern and one of the most pressing public health challenges worldwide. The prevalence of overweight and obesity rose globally, regionally, and across all nations between 1990 and 2021. By 2021, an estimated 2.11 billion (95% UI: 2.09–2.13) adults aged 25 and older were affected by overweight and obesity, accounting for nearly half of the global adult population (45.1% [44.7–45.4]). If historical trends continue, projections suggest that by 2050, the number of adults living with overweight and obesity will increase to 3.80 billion (95% UI: 3.39–4.04), surpassing half of the anticipated global adult population [1]. Obesity is responsible for over four million deaths annually and significantly increases morbidity. The disease not only reduces life expectancy but also severely impacts quality of life [2]. Economically, the burden of obesity is staggering, encompassing direct medical expenses as well as indirect costs such as lost productivity [3].

The increasing prevalence of obesity underscores the urgent need for coordinated action. Interventions at the individual, community, and policy levels are essential to reversing the obesity epidemic and mitigating its impact on society [4]. Obesity management requires a multifaceted approach primarily involving lifestyle changes, including dietary adjustments, increased physical activity, and behavioral therapy. These interventions are considered the principal strategies for managing obesity, and when combined, they are the most effective way to achieve weight loss and maintain a healthy weight over the long term [5].

Traditional obesity-management approaches face several limitations that digital health interventions may uniquely address. Scalability constraints limit the reach of intensive face-to-face interventions, with healthcare systems unable to provide counseling at an adequate frequency for the growing obese population. Accessibility barriers including geographic distance, scheduling conflicts, and socioeconomic factors prevent many patients from accessing specialized obesity care. Limited real-time monitoring in traditional approaches means that behavioral lapses or weight regain often go undetected until scheduled appointments, missing critical intervention windows. Digital health interventions address these gaps through continuous accessibility (24/7 availability), real-time feedback mechanisms that can provide immediate behavioral reinforcement, cost-efficiency that enables broader population reach, and objective monitoring capabilities that operate through integrated sensors and self-reporting platforms. Furthermore, digital tools offer personalization at scale through algorithmic adaptation to individual progress patterns, potentially overcoming the “one-size-fits-all” limitation of traditional group-based interventions.

Although they are effective, current treatment options have not yet succeeded in reversing the upward trend in adult overweight and obesity rates. No country has managed to halt the rising prevalence of obesity. Without swift and effective action, these rates are expected to continue climbing worldwide [1].

Advancements in mobile apps, telemedicine, and wearable devices now provide an opportunity to enhance patient engagement, monitor progress in real time, and support behavior change [6]. E-health and m-health, which are integral components of digital health focused on technology use, hold significant potential for improving obesity management. These approaches may offer substantial opportunities to enhance obesity care by providing accessible, personalized, and continuous management [7].

The aim of this systematic review is to evaluate whether digital health interventions (e-Health and m-Health) result in changes in BMI/body weight and/or improvements in health outcomes or complications by comparing trial groups to control groups.

## 2. Materials and Methods

### 2.1. Protocol Registration

The systematic review was conducted in accordance with the Preferred Reporting Items for Systematic Reviews and Meta-Analyses 2020 guidelines [8] and was registered in the PROSPERO database (number: CRD42024559942; https://www.crd.york.ac.uk/prospero/display_record.php?ID=CRD42024559942, accessed on 19 June 2024).

### 2.2. Search Strategy and Data Sources

A comprehensive literature search was conducted, including works published from the inception of the databases up to 4 October 2024. The Medline, Scopus, and Web of Science databases were used to identify original articles that investigated the association between E-health OR M-health and obesity management in adults. The search strategy, including medical subject headings (MeSH) and keywords, is outlined in the Appendix A. The search was limited to human studies and articles published in English.

### 2.3. Study Selection

A two-step process was used to select eligible articles after duplicates had been removed, as follows:Title and Abstract Screening: Initial screening was performed based on the titles and abstracts of the articles.Full-Text Review: Full texts of potentially eligible studies were obtained and assessed for final inclusion.

Two investigators independently carried out the selection process. Any disagreements that arose during the selection were resolved through discussion or by consulting a third author.

Articles were included if they met the following criteria:Evaluated digital health interventions (e-Health and m-Health), either as standalone interventions or as part of mixed approaches combining digital components with limited in-person elements, compared to no treatment or traditional management approaches providing educational materials.Focused on overweight or obese adults aged over 18 years with BMIs above 25 kg/m^2^.Utilized a randomized controlled trial (RCT) design that compared two or more groups, one of which received an e-Health or m-Health intervention.Reported health endpoints, including anthropometric endpoints (BMI, weight, waist circumference, and body-fat percentage) and lifestyle behavior outcomes (physical activity, energy intake and eating behaviors) for individuals enrolled in the study.

The following studies were excluded: single-arm or open-label RCTs, study protocols, reviews, conference abstracts, and publications describing the effects of digital health interventions on infants or the elderly.

### 2.4. Data Extraction and Synthesis

Information extracted from each study included the first author’s last name, year of publication, trial status, country, study population (sex, age, and risk behaviors), number of participants and characteristics of participants in both the experimental and control arms, type of intervention and control, study endpoints with definitions, and results for each endpoint. Two investigators independently carried out the data extraction. A narrative synthesis of the results was planned due to the expected heterogeneity in terms of patient type, intervention type, endpoints, and the timing of endpoint assessments.

### 2.5. Quality Assessment

The quality of included RCTs was assessed using the Cochrane Risk of Bias Tool (RoB 2) [9]. Reviewers assessed the risk of bias independently and in duplicate. For each of the following items, a judgment of “low risk” or “probably low risk” (indicating bias is not present or is unlikely to alter findings) or “high risk” or “probably high risk” (indicating bias may alter the results) was made: sequence generation, allocation concealment, blinding of participants and clinicians, blinding of outcome assessment, blinding of data collectors and data analysts, loss to follow-up, and other factors (e.g., trial stopped early). The overall risk of bias for each included trial was considered low if it was evaluated as being low or probably low risk across all domains or high if it was evaluated as high or probably high risk in one or more domains. Disagreements were resolved through discussion or, if necessary, by consultation with a third reviewer.

### 2.6. Statistical Analysis

#### Strategy for Data Synthesis

In accordance with PRISMA guidelines [8], a minimum of two studies was required for inclusion in any meta-analysis. Synthesized data comprised (i) mean differences in changes in anthropometric indicators and (ii) differences in lifestyle-related outcomes between intervention and control groups, as measured using standardized mean differences (Cohen’s d), where appropriate.

Statistical heterogeneity was assessed using Cochran’s Q and the *I*^2^ statistic. *I*^2^ values of <25%, 25–50%, and >50% were interpreted as indicating low, moderate, and high heterogeneity, respectively. A *p*-value < 0.05 for the Q statistic was considered indicative of significant heterogeneity [10]. In the presence of significant heterogeneity, a random-effects model was applied; otherwise, a fixed-effects model was used. All analyses were performed using ProMeta software (version 3), and statistical significance was set at *p* < 0.05.

## 3. Results

### 3.1. Study Selection

The systematic search identified 230 records, with 37 from Medline, 101 from Scopus, and 92 from Web of Science. When duplicates were removed, 98 records were excluded. A total of 132 records were screened by title and abstract, and 31 of these were assessed for eligibility. Nine records were excluded after full-text review, and, ultimately, 22 studies were deemed eligible for inclusion in the review [1,2,3,4,5,6,7,8,9,10,11,12,13,14,15,16,17,18,19,20,21,22] (Figure 1).

### 3.2. Results of Systematic Review

Table 1 summarizes the characteristics of the 22 articles included in this study. The studies were conducted between 2013 and 2023 in 12 countries (USA, Canada, Portugal, Spain, England, Belgium, Germany, Austria, India, South Korea, Korea, and Australia).

Eight studies focused on obese adults [1,2,3,8,10,13,17,18]. One study focused on adults with moderate overweight [16]; two studies included the general population [12], one study examined a population at risk of cardiovascular disease [12]; one study focused on adults diagnosed with binge-eating disorder (BED) [19]; and one study investigated postpartum women [6].

Eighteen studies were two-arm trials [1,2,3,4,5,6,7,8,9,10,11,13,14,15,16,17,19,20,22], while three studies had three arms [7,11,13]. Three studies were prospective cohort studies [12,18]; two were randomized controlled trials [12,16]; and one was an intervention-development study with a 13-week follow-up [21].

Two articles described interventions using biofeedback devices [1,2]; four articles described interventions using smartphone-based coaching apps [3,7,11,15]; one used a daily text-messaging intervention [22]; eight articles described various e-coaching tools [9,10,12,13,19,21]; one used a DVD-based intervention [20]; one used an online tutorial [8]; and four utilized web-based interventions [4,5,16]. Two articles involved mixed e-health tools [6,18].

Regarding outcomes, twenty articles focused on anthropometric measures, such as BMI, weight, waist circumference, and body composition (body fat %) [1,8,9,10,11,13,14,15,16,17,18,19,20], and eleven studies addressed eating and/or exercise behaviors [4,5,6,7,11,13,14,18,19,21,22].

The risk-of-bias assessment, based on the five domains of the RoB 2.0 tool (randomization process, deviations from intended interventions, missing outcome data, measurement of outcomes, and selection of the reported result), revealed a predominance of judgments categorized as “Some Concerns” and “High Risk,” indicating an overall methodological quality that is heterogeneous and, in several cases, critically limited across the included studies (Appendix A).

### 3.3. Results of GRADE


**Outcome**

**Certainty**

**Reasons for Downgrade**

**BMI**
LowHigh inconsistency, imprecision
**Weight**
ModerateImprecision
**Waist circumference**
ModerateInconsistency
**Body Fat %**
LowInconsistency, imprecision
**Physical activity**
LowHigh inconsistency, imprecision
**Energy intake**
Very LowVery high inconsistency, imprecision, few studies
**Eating behavior**
LowHigh inconsistency, imprecision

### 3.4. Results of Metanalysis

#### 3.4.1. Anthropometric Indicators

The meta-analysis did not reveal significant differences in anthropometric outcomes based on population characteristics such as sex, age, or population type. Specifically, the exclusion of studies involving participants with pre-existing medical conditions at baseline or those who were pregnant or in the postpartum period, as well as variations in the type of obesity-management intervention (e-Health vs. m-Health), did not significantly alter the results, as shown in Table 2.

The meta-analysis showed no significant changes in most anthropometric outcomes (Table 2). For BMI, the pooled weighted mean difference (WMD) was −0.43 (95% CI: −1.15 to 0.30; *p* = 0.247), indicating no significant effect. Similarly, body-weight analysis based on 10 studies yielded a non-significant WMD of 0.42 (95% CI: −0.45 to 1.29; *p* = 0.341). Waist circumference, however, showed a statistically significant reduction, with a WMD of −1.77 (95% CI: −3.10 to −0.44; *p* = 0.009), suggesting a beneficial effect. For body-fat percentage, the analysis revealed a near-significant reduction (WMD = −0.79; 95% CI: −1.63 to 0.06; *p* = 0.068), suggesting a trend toward improvement but not reaching conventional statistical significance.

Highly significant heterogeneity was observed in several anthropometric outcomes (Table 2), including BMI, waist circumference, and body-fat percentage, indicating variability across studies.

In the sensitivity analysis excluding studies with high risk of bias (D 1 and D2 of Robs tool 2), the overall effect size remained consistent (WMD: −0.12, 95% CI −0.24–0.01), suggesting the robustness of the findings. Sensitivity analysis indicated that the exclusion of the study by Duncan et al. [22] yielded a statistically significant overall effect in anthropometric indicators (Cohen’s d = −0.13; 95% CI: −0.26 to −0.01; *p* = 0.035). A similar result was observed upon exclusion of the study by Hutchesson et al. [24], with an overall effect size of Cohen’s d = −0.13 (95% CI: −0.25 to 0.00; *p* = 0.047), suggesting that both studies exerted a considerable influence on the heterogeneity and statistical significance of the pooled estimates.

#### 3.4.2. Lifestyle Behaviors

While no statistically significant differences in lifestyle-behavior change were observed across sex, population type, or intervention modality, subgroup analysis revealed that individuals belonging to Generation X (i.e., those born after 1965) demonstrated a statistically significant improvement in health-related behaviors in response to e-Health and m-Health interventions (*p* < 0.05), suggesting a greater responsiveness of this cohort to digital health strategies (Table 2).

No significant improvements were observed for specific lifestyle-behavior outcomes (Table 2). For physical activity, the pooled standardized mean difference (SMD) was −0.01 (95% CI: −0.39 to 0.36; *p* = 0.939), indicating a negligible effect. Energy intake showed a borderline-significant reduction (SMD = −0.93; 95% CI: −1.88 to 0.01; *p* = 0.052), suggesting a potential trend. In contrast, eating behavior showed no significant change, with an SMD of −0.13 (95% CI: −0.40 to 0.14; *p* = 0.341).

All lifestyle-behavior outcomes exhibited highly significant heterogeneity, suggesting a wide range of effects across the studies included in these analyses (Table 2).

In the sensitivity analysis excluding studies with high risk of bias (D 1 and D2 of Robs tool 2), the overall effect size remained consistent (Cohen’s d: −0.01, 95% CI −0.16–0.15), suggesting the robustness of the findings. Sensitivity analysis showed that excluding Johnson et al. [23] (Cohen’s d = −0.24; 95% CI: −0.44 to −0.03; *p* = 0.023) or Wang et al. [32] (Cohen’s d = −0.26; 95% CI: −0.46 to −0.06; *p* = 0.012) yielded statistically significant effects, indicating their substantial impact on heterogeneity and overall significance.

#### 3.4.3. Publication Bias

No significant publication bias was detected by either Egger’s or Begg’s method, as shown by the *p* values (Table 2) and funnel plots (Appendix A—Forest plot and Funnel linked to anthropometric outcome and behavior outcome).

## 4. Discussion

This meta-analysis of 22 randomized controlled trials evaluated the effectiveness of digital health interventions (e-Health and m-Health) in obesity management among adults. The results demonstrate a mixed pattern of effectiveness, with limited but statistically significant benefits for specific anthropometric outcomes. In summary, digital interventions produced a significant reduction in waist circumference (−1.77 cm) but showed no significant effects on BMI, body weight, or lifestyle-related behaviors.

In fact, the only anthropometric outcome that reached statistical significance was waist circumference, with a mean reduction of 1.77 cm (95% CI: −3.10 to −0.44; *p* = 0.009). This result is clinically relevant. considering that waist circumference is an independent predictor of cardiovascular and metabolic risk. The 1.77 cm reduction in waist circumference, while modest in absolute terms, has substantial clinical significance. According to current cardiovascular-risk-stratification guidelines, reductions of 1–2 cm in waist circumference are associated with meaningful decreases in metabolic risk factors. Evidence from large epidemiological studies demonstrates that even modest reductions in waist circumference correlate with decreased cardiovascular disease risk independently of BMI changes. Regarding BMI and body weight, digital interventions showed no significant efficacy, with WMD of −0.43 kg/m^2^ (*p* = 0.247) and 0.42 kg (*p* = 0.341), respectively. Body-fat percentage showed a trend toward improvement (−0.79%; *p* = 0.068) that, while not reaching conventional significance, suggests a potential biologically plausible effect of digital interventions on body composition.

Although the meta-analysis did not detect statistically significant differences in lifestyle behavior change across sex, population type, or intervention modality, subgroup analysis indicated that Generation X participants (i.e., born after 1965) exhibited a statistically significant improvement in health-related behaviors in response to both e-Health and m-Health interventions (*p* < 0.05). This finding aligns with existing literature documenting Generation X’s greater readiness to adopt digital health technologies and derive health benefits from them. This finding is consistent with existing literature that underscores Generation X’s greater readiness to adopt digital health technologies and benefit from them. As noted by Papp-Zipernovszky et al. [33], while online health information-seeking behavior is relatively similar across generations, eHealth literacy is significantly lower among Baby Boomers (born before 1965). Notably, it is only from Generation X onward that a markedly higher capacity to engage with and derive benefit from digital health interventions becomes evident.

Analysis of the primary data reveals interesting patterns in behavioral outcomes. For physical activity, studies such as Ruiz-Cortes et al. [14] and Múzquiz-Barberá et al. [15] showed increases in METs (from ~2700–3000 baseline to ~2900–3400 post-intervention), while Bijlholt et al. [16] reported increases in METs and reductions in sedentary time. However, the marked methodological heterogeneity (*I*^2^ = 86.62%) indicates that these effects are not consistent across studies.

For energy intake, the borderline-significant trend toward reduction (SMD = −0.93; *p* = 0.052) is supported by specific evidence: Bijlholt et al. [16] showed reductions from 1409 to 1204 kcal/day in the intervention group, while Fenton et al. [17] reported decreases in energy intake in both intervention subgroups (traditional and enhanced). However, the high heterogeneity (*I*^2^ = 93.63%) suggests that effectiveness varies considerably among types of digital interventions.

For eating behaviors, the analysis included various domains of the Dutch Eating Behavior Questionnaire (DEBQ), with Ruiz-Cortes et al. [14] measuring emotional, restrictive, and external behaviors and Bijlholt et al. [16] focusing on restrictive behaviors. The absence of significant effects (SMD = −0.13; *p* = 0.341) may reflect the complexity of modifying established behavioral patterns through digital interventions, in addition to variability in the instruments used.

The modest but significant effect on waist circumference suggests that digital interventions may be valuable for addressing the scalability crisis in obesity management, which arises because traditional intensive interventions cannot reach the millions requiring treatment. The potential of digital tools as cost-effective interventions becomes crucial when considering population-level implementation, as even small individual effects can translate into substantial public health benefits when the tools are applied broadly.

A critical finding emerging from this meta-analysis is the high heterogeneity observed in most analyzed outcomes (*I*^2^ > 80% for BMI, physical activity, energy intake, and eating behavior). Detailed analysis of primary studies reveals multiple sources of variability that explain this heterogeneity, such as the following:Diversity in sample sizes, which varied dramatically from 14 participants (Fenton et al. [17]—enhanced intervention subgroup) to over 1000 participants (Bijlholt et al. [16], with 487 controls and 533 interventions), and target populations, which included obese adults, overweight individuals, postpartum women, patients with eating disorders, and populations with high cardiovascular risk. The variable dropout rate between studies (e.g., Bijlholt et al. [16] lost approximately 30% of participants during follow-up) represents an additional source of heterogeneity.Variety in measurement instruments. For example, for physical activity, some studies used METs via IPAQ (Ruiz-Cortes et al. [14], Múzquiz-Barberá et al. [15]), while others measured sedentary time in minutes/day. For energy intake, units included both kcal/day and kJ/day, and values were measured using different assessment methodologies. For eating behaviors, different questionnaires (Dutch Eating Behavior Questionnaire DEBQ, FFQ) with variable scoring scales were used.Heterogeneity of digital interventions: the range of technologies used includes biofeedback devices (Bernardo et al. [11], Choi et al. [12]), smartphone apps with coaching (Domal et al. [13], Fenton et al. [17], Duncan et al. [22], Lee et al. [6]), e-coaching systems (Yu et al. [19], Alencar et al. [20], Johnson et al. [23]), web-based interventions (Ruiz-Cortes et al. [14], Múzquiz-Barberá et al. [15]), and mixed approaches. Each technology presents significantly different mechanisms of action, levels of interactivity, and frequencies of use. Additionally, the inclusion of both purely digital interventions and mixed approaches (combining digital tools with limited in-person components) contributed to the observed heterogeneity, as the hybrid nature of some interventions may have enhanced their effectiveness through mechanisms that differ from those of standalone digital tools.

The GRADE assessment reveals that even our most promising finding (reduction in waist circumference) represents only moderate-certainty evidence, emphasizing the need for cautious interpretation of results. The predominance of low- to very-low-certainty evidence across behavioral outcomes highlights significant limitations in the current evidence base for digital health interventions in obesity management.

Our results are consistent with previous systematic reviews that have evidenced modest effects of digital interventions on weight loss. A meta-analysis by Sorgente et al. (2017) [34] reported body-weight reductions of 2.56 kg in app-based interventions, while Flores Mateo et al. (2015) [35] observed BMI reductions of 0.43 kg/m^2^. Our results, showing non-significant effects on weight and BMI, may suggest that the effectiveness of digital interventions may be more limited than initially hypothesized. However, our study distinguishes itself by identifying a significant effect on waist circumference, an outcome that is often overlooked in previous meta-analyses but that is of great clinical relevance; in fact, recent guidelines from the American Heart Association (2021) and the European Society of Cardiology (2021) have emphasized the importance of waist circumference as a cardiovascular risk marker independent of BMI, making our result particularly relevant to clinical practice [36,37].

The absence of significant effects on BMI, body weight, and behavioral outcomes may also be explained by several methodological and intervention-related factors. Short intervention durations represented a critical limitation, with most studies (68%) having follow-up periods ≤ 6 months, a duration insufficient for detecting substantial weight changes or establishing sustained behavioral modifications. Poor adherence and engagement were evident in several studies, with dropout rates reaching 30% (Bijlholt et al. [16]), suggesting that digital interventions may face unique challenges in maintaining long-term participant engagement compared to traditional face-to-face approaches. Variability in intervention intensity and mechanisms likely diluted observable effects. Study tools ranged from simple text messaging (Wang et al. [32]) to complex multi-component platforms (Hutchesson et al. [24]), each targeting different behavioral pathways and having varying degrees of personalization and feedback. Self-reporting bias in behavioral outcomes may have further obscured true intervention effects, as participants could overestimate adherence to physical-activity or dietary changes, particularly in digital formats in which objective monitoring was limited.

Two further factors that influenced measures and help to explain the non-significance of the results are intervention fidelity and digital literacy. Intervention fidelity varied substantially, with some studies reporting minimal monitoring of actual app usage or engagement levels. For instance, studies utilizing complex multicomponent platforms (Hutchesson et al. [24]) may have suffered from poor fidelity if participants used only selected features, while simpler interventions (Wang et al. [32] text messaging) likely achieved higher fidelity but lower potential impact. Disparities in digital literacy among participants likely created differential intervention effects that contributed to high heterogeneity. Older adults or those with limited technology experience may have experienced reduced benefits regardless of intervention quality. Most studies failed to assess baseline digital competency or provide adequate technology orientation, potentially undermining intervention effectiveness in digitally naive populations. Future digital health interventions must incorporate digital-literacy assessments and targeted training to optimize engagement and ensure equitable access to intervention benefits.

The observed heterogeneity and the factors analyzed here suggest the need for a more personalized digital health intervention, possibly with multi-level personalization through dynamic adaptive-feedback systems or through integrated clinician–digital hybrid models. For example, rather than static goal-setting, future interventions should implement real-time adaptive feedback based on continuous monitoring data; alternatively, algorithmic triage systems could be used to identify participants requiring human clinical intervention based on predetermined risk markers (lack of progress, concerning behavioral patterns, medical complications). Artificial intelligence could be a support in analyzing individual response patterns to adjust intervention intensity, timing, or modalities.

Finally, the results of this meta-analysis suggest that digital health interventions should be considered as complementary rather than substitutive tools in the multimodal approach to obesity management. The significant effect on waist circumference, though modest in absolute terms, could translate into clinically relevant benefits if it were integrated into a broader therapeutic pathway. The high heterogeneity observed and specific patterns emerging from primary data analysis suggest that the effectiveness of digital interventions depends significantly on the intervention and the characteristics of the target population. Clinicians should consider that more intensive interventions (such as multicomponent interventions) might be more effective but require greater support to maintain adherence.

## 5. Study Limitations

Methodological quality of primary studies: quality assessment using the RoB 2.0 tool revealed a predominance of judgments categorized as “Some Concerns” and “High Risk” across the five evaluated domains (randomization process, deviations from intended interventions, missing outcome data, outcome measurement, and selection of reported results). Particular concern arises from the risk of bias related to participant blinding, which is inherently difficult to implement in digital behavioral interventions, and from management of missing data, a problem highlighted by the high dropout rates in some studies.

Methodological heterogeneity: as discussed, the high heterogeneity significantly limits our ability to draw definitive conclusions about the overall effectiveness of digital interventions.

Follow-up duration and sustainability: most included studies present short-to-medium-term follow-up, limiting our ability to assess the long-term sustainability of the observed effects.

Subgroup analyses based on participant age, obesity grade, type and duration of intervention, and risk of bias (high vs. low) were not planned.

## 6. Conclusions

This meta-analysis provides limited but encouraging evidence for the effectiveness of digital health interventions. Such interventions may offer modest but specific benefits, specifically for reduction in waist circumference in overweight and obese adults, while effects on other anthropometric measures and behavioral parameters remain undemonstrated. The high heterogeneity observed underscores the need for more personalized approaches and future research focused on identifying the most effective components of digital interventions. While digital technologies represent promising tools in the therapeutic armamentarium for obesity, our results suggest that their implementation should occur within the context of integrated and personalized approaches to weight management.

## Figures and Tables

**Figure 1 nutrients-17-02200-f001:**
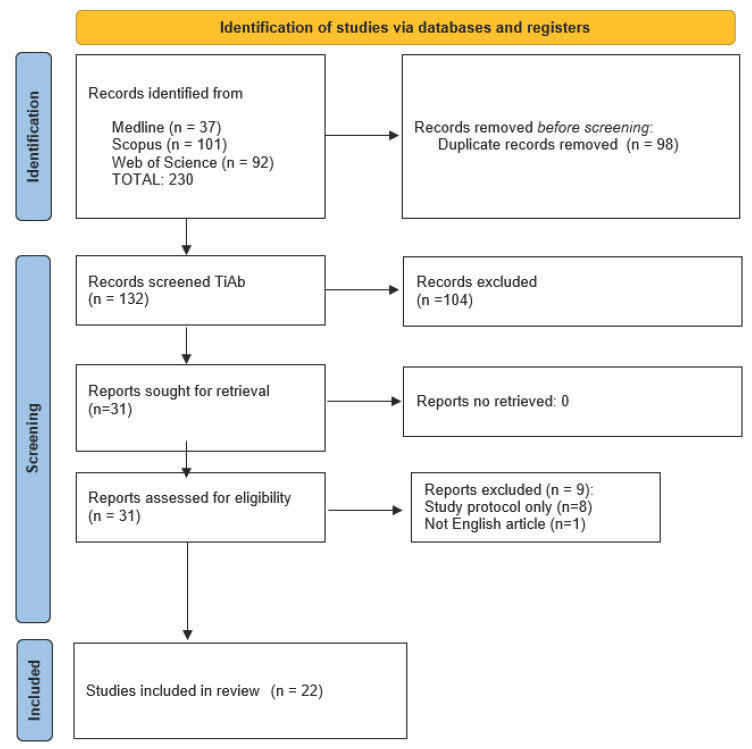
PRISMA Flow chart.

**Table 1 nutrients-17-02200-t001:** Study characteristics.

Author; Year; Country	Population; Sample; Mean Age (SD); Male Population %	Targeted Risk Behaviors	Study Design; Follow Up; Frequency; Assessment	Intervention	Intervention Components	Comparison Group	Endpoints
Bernardo et al., 2024, Portugal [11]	Obese; n = 24; age ≥ 18 y; 0% males	Pregnant between the 6th and 20th gestational week	Two-arm trial Follow-up: 8 weeks	**e-Health** e-Health: Remote exercise program delivered by a Phoenix^®^ biofeedback device	n = 12 Phoenix^®^ Device available 24 h/7 days per week; pregnant women were encouraged to exercise at least three times a week.	n = 12 Standard care	**Anthropometric measures** Weight
Choi et al., 2023, Korea [12]	Obese; n = 30; age: 39 ± 11 y; 0% males	Body fat > 30%	Two-arm trial Follow-up: 12 weeks Muscle function almost two times/week, cardiorespiratory endurance and flexibility almost five times/week	**m-Health** m-Health: Fitbit Charge + Fitbit app linked to the AI Fit webpage	n = 15 Combined exercise (muscle function, cardiorespiratory endurance, flexibility); the exercise programs were followed while watching videos, and the participants were able to obtain feedback on exercise performance through a mobile chat program linked to AI fit.	n = 15 Normal daily routines	**Anthropometric measures** Weight BMI Body fat Waist circumference
Domal et al., 2023, India [13]	Obese; n = 20; age: 18–35 y; 50% males	Men, waist circumference >90 cm; women, waist circumference >80 cm	Two-arm trial Follow-up: 8 weeks	**m-Health** Smartphone-based application (EXi app.)	n = 10 Video conferencing	n = 10 Standard care	**Anthropometric measures** Weight BMI Body fat Waist circumference
Riuz-Cortes et al., 2023, Spain [14]	Overweight/obese; n = 132; age: 46–67 y; 55% males	Hypertension	Two-arm trial Follow-up: 12 weeks	**e-Health** eHealth: Living Better web-based program and web page (Wix)	n = 70 Psychological strategies that encourage a healthy lifestyle, healthy eating habits and increased PA, with the participants’ own doctor appearing in the audiovisual material	n = 62 Standard protocol with unknown doctor	**Anthropometric measures** BMI **Behaviors** Adherence to Mediterranean diet Eating behaviors Physical activity
Múzquiz-Barberá et al., 2023, Spain [15]	Overweight/obese; n = 132; age: 46–67 y; 55% males	Hypertension	Two-arm trial Follow-up: 12 weeks	**e-Health** eHealth: Living Better web-based program and web page (Wix)	n = 70 Psychological strategies that encourage a healthy lifestyle, healthy eating habits, and increased PA, with the participants’ own doctor appearing in the audiovisual material	n = 62 Standard protocol with unknown doctor	**Anthropometric measures** BMI **Behaviors** Physical activity
Bijlholt et al., 2021, Belgium [16]	Overweight/obese; n = 1450; age: ≥ 18y; 0% males	Women in the postpartum period	Two-arm trial Follow-up: 1 year Assessment at T1 6 months and T2 12 months after delivery	**e-Health** INTER-ACT e-Health	n = 193 OW; n = 89 OB	Standard care n = 181 OW and 74 OB	**Behaviors** Eating behaviors Energy intake Physical activity
Fenton et al., 2021, Australia [17]	Overweight/obese; n = 116; age: 19–65 y (44.5 ± 10.5); 30% males	BMI 31.7 kg/m^2^	Three arms Follow-up: 1 year Assessment at T1 6 months and T2 12 months EG m-Health (pooled Enhanced + Traditional) two arms/CG one arm	***m-Health*** Smartphone app providing educational materials, goal-setting, self-monitoring, and feedback and one face-to-face dietary consultation, a Fitbit, and scales.	m-Health: smartphone app n = 39 Enhanced-intervention group (provided with stress-management and relaxation techniques and received access to a sleep intervention via the app); n = 41 Traditional-intervention group (provided with stress-management and relaxation techniques but no sleep intervention)	n = 36 Standard care	**Anthropometric measures** Weight BMI Waist circumference (only at baseline!) **Behaviors** Energy intake Physical activity
Welzel et al., 2021, Germany [18]	Obese; n = 135; age: 43.3 ± 10.7 y; 38% males	BMI = 39 ± 6.0 kg/m^2^	Two-arm trial Follow-up: 1 year Assessment at T1 6 months T2 12 months	***e-Health*** Online tutorial	n = 65 Access to 5As of Obesity Management tutorial	n = 70 Standard care	**Anthropometric measures** Weight BMI
Yu et al., 2021, USA [19]	Overweight/obese; n = 13; age: 20–73 y; 0% males	Binge-eating disorder	Two arms Follow-up: 3 months	** *m-Health* **	n = 4 Videoconferencing-based treatment program	n = 9 Standard care	**Anthropometric measures** Weight BMI ***Weight in NO Disorder***
Alencar et al., 2020, USA [20]	Obese; n = 25 (M and F); age: 41.5 ± 13.6 y; NR% males	BMI = 34.6 ± 4.33 kg/m^2^	Two arms Follow-up: 12 weeks	***m-Health*** Two wireless devices that connected them directly with the research team	n = 13 Health video-coaching; telehealth video conferencing for health coaching with a dietitian and a physician	n = 12 Standard care	**Behaviors** Adherence to wireless devices;
							**Anthropometric measures** Weight (Weight Week Loss)
Yousuful et al., 2019; HAPPY NL, HAPPY AZM Netherlands [21]			Prospective cohort study NO RCT				
Yousuful et al., 2019; HAPPY LONDON, England [21]	Overweight/obese; n = 402; age: 65.5 (40–74 y); 67% males	BMI E-coaching: 27.4 kg/m^2^ Standard: 27.1 kg/m^2^ a 10-year CVD risk score of ≥10%; excluded people with an established cardiovascular disease	Single RCT Follow-up: 6 months	***e-Health*** e-Coaching MyCLIC: e-Coaching lifestyle intervention tool	n = ? e-Coaching received 6 months of tailored advice and were able to enter their MyCLIC pages with personal logins.	n = ? Standard care	**Anthropometric measures** Weight BMI
Duncan et al., 2020, Australia [22]	Overweight/obese; n = 116; age: 19–65 y (44.5 ± 10.5); 30% males	BMI 31.7 kg/m^2^	Three arms Follow-up: 1 year Assessment at T1 6 months, T2 12 months EG m-Health (pooled Enhanced + Traditional) 2 arms/CG 1 arm	***m-Health*** Smartphone app providing educational materials, goal-setting, self-monitoring and feedback, and also included one face-to-face dietary consultation, a Fitbit and scales.	n = 39 Enhanced-intervention group (received access to a sleep intervention via the app and participant handbook targeted a reduction in sleep timing variability, promoted sleep hygiene behaviors and provided stress management and relaxation techniques. n = 41 Traditional-intervention group (no sleep intervention)	n = 36 Standard care	**Anthropometric measures** Body weight BMI Waist circumference **Behaviors** Energy intake Physical activity
Johnson et al., 2019, USA [23]	Obese; n = 30 (M and F); age: 32–46 y; NR% males		Three arms Follow-up: 12 weeks	***e-Health*** Telemedicine-based health coaching	n = 10 Video conference telemedicine-based health coaching	n = 10 In-person group n = 10 Standard care	**Anthropometric measures** Weight BMI **Behaviors** Physical activity
Hutchesson et al., 2018, Australia [24]	Overweight/obese; n = 57; age: 18 ÷ 35 y (27.1 ± 4.7); 0% males	(56.1% overweight and 43.9% obese)	Two arms Follow-up: 6 months	***e-Health*** e-Health: Multicomponent intervention consisting of five delivery modes: website, social media, smartphone application, email, text messages.	n = 29	n = 28 Normal daily routines	**Anthropometric measures** Weight BMI Waist circumference Body fat **Behaviors** Energy intake Physical activity
Lee et al., 2018, South Korea [25]	Overweight/obese; n = 324; age: > 20 y; 47% males	BMI ≥ 25 kg/m^2^; metabolic syndrome	Two-arm trial Follow-up: 24 weeks	***m-Health*** Smartphones equipped with the SmartCare application and a Bioimpedance Analyzer via Bluetooth to facilitate telemonitoring.	n = 177	n = 147 Standard care	**Anthropometric measures** Weight BMI Waist circumference Body fat
Melchart et al., 2017, Germany [26]	Overweight/obese; n = 166; age: IHM: 49.9 (9.7) y, UC: 52.2 (19) y; IHM 25,2% males, UC: 27.3% males	Excluded pregnant people with diabetes, hypertension grade 2, disease	Two arms Follow-up: 1 year	***e-Health*** Individual Health Management (IHM): lifestyle-modification program with a 3 month reduction phase and 9 month maintenance phase. The program encompasses access to a web-based health portal providing advice regarding food, exercise, and relaxation and allows personalized feedback for control of the progress.	n = 109	n = 57 Standard care	**Anthropometric measures** Weight BMI Waist circumference
Rader et al., 2017, Austria [27]	Obese; n = 84; age: 18–80 y; 0% males		Two-arm trial Follow-up: 1 year Assessment at T1 6 months T2 12 months	***e-Health*** Web-based intervention (WBI); e-Health web-based intervention (WBI) conducted subsequent to an initial face-to-face lifestyle treatment, with an introductory phase (4 months) and a training phase (2 months) in which each group was trained in using the appropriate instrument according to their study arm. In the following 6 months, participants used either the WBI or the PMI for follow-up support.	n = 21	n = 22 Standard care	**Anthropometric measures** BMI
Rogers et al.; 2016; USA [28]	Obese; n = 39; age: 39.9 ± 11.5 y; 79.5% males	Sedentary adults, BMI 39.5 ± 2.8 kg m^2^, excluded pregnant women, people taking medications or with chronical disease	Three arms Follow-up: 6 months Assessments at T1 3 months T2 6 months	***e-Health*** Technology-based intervention combined with a monthly intervention telephone call (TECH); participants were provided with the BodyMedia^®^ FIT System, a wearable device. Enhanced technology-based system combined with a monthly intervention telephone call (EN-TECH): BodyMedia^®^ FIT System with the LINK activity monitor	n = 12 TECH; n = 13 EN-TECH	n = 14 Standard in-person group-based behavioral weight-loss intervention	**Anthropometric measures** Weight BMI Waist circumference Body fat **Behaviors** Eating behaviors Energy intake Physical activity
Wagner et al., 2016; Germany [29]	Overweight/obese; n 139; age: 35.1 (9.9) y; 3.6% males	Binge-eating disorder Excluded if BN or anorexia nervosa, severe major depression, acute suicidal symptoms, abuse, medical condition influencing weigh	Two arms Follow-up: 1 year Assessments at T1 3 months T2 6 months T3 12 months	***e-Health*** 16-week intervention Internet-based cognitive–behavioral intervention or a wait-list condition. Internet-based cognitive–behavioral intervention Guided therapy with intensive therapist contact, 11 personalized structured writing assignments and individualized feedback from trained therapists, complimentary daily eating and activity diaries, week plans, and psychoeducation as applied in cognitive–behavioral treatments.	n = 69	n = 70	**Anthropometric measures** Weight BMI **Behaviors** Eating behaviors
Gregosky et al., 2015, USA [30]	Overweight/obese; n = 54; age: 37.7 ± 9.9 y; 7% males	BMI: 30.8 ± 8.6	Two-arm trial Follow-up: 10 weeks;	***e-Health*** e-Health: take home DVD. A prize of $15 was awarded to those who lost ≥ 15 lbs.	n = 45 Physical-activity program on DVD	n = 9 Standard care	**Anthropometric measures** Weight BMI
Tomkins Lane et al., 2015, Canada [31]	Overweight/obese; n= 10 age: 67.5 ± 6.7 y; 40% males	Lumbar spinal stenosis Excluded people with comorbidities that limit walking.	Intervention development and pilot Follow-up: 13 weeks	***e-Health*** Lifestyle-modification approach of physical activity and nutrition education, delivered through an e-health platform. Participants received a pedometer and a personalized consultation with a dietitian and an exercise physiologist. For 12 weeks, participants logged on to the e-health Web site to access personal step goals, nutrition-education videos, and a discussion board.	n = ?	n = ?	**Anthropometric measures** Weight Waist circumference
Wang et al., 2015, USA [32]	Overweight/obese; n = 67; age: 48.2 (11.7) y; 9% males	Not meeting recommended levels of physical activity (PA < 150 min/week of MVPA, 48 ability to safely increase PA)	Two-arm trial Follow-up: 6 weeks Assessment weekly Population wore a Fitbit One. The Fitbit One included a wearable tracker for instant feedback on performance and a website/mobile application (app) for detailed summaries.	***e-Health*** Test the effects of daily text messaging as simple prompts to increase PA. Three daily SMS-based PAprompts.	n = 33	n = 34; Self-monitoring with Fitbit One only	**Behaviors** Physical activity

Overweight: BMI ≥ 25 kg/m^2^. Obese: BMI ≥ 30 kg/m^2^. Standard care for obesity management: face-to-face consultation, written information with advice for healthy behaviors, printed manual with advice for healthy behaviors, group-based treatment program but NO M-Health or E-Health. Normal daily routines: usual eating and physical-activity habits. Body fat: body fat (%), body fat (kg), fat mass, and fat-free mass. Eating behaviors: uncontrolled eating, restrained eating, emotional eating, food intake, macronutrient/mineral intake, and fried/takeaway foods. Energy intake: total energy intake (kJ/d); food intake, macronutrient/mineral intake, and fried/takeaway foods. Physical activity: level (METs-min/week), activity/sedentary time, maximum continuous activity, intensity level (minutes/week), and steps (n/day).

**Table 2 nutrients-17-02200-t002:** Results of stratified analysis according to the outcomes identified.

			WMD		Test of Heterogeneity			Publication Bias	
	Studies	Effect-Sizes	Value (95% CI)	*p*	Q	*I*^2^%	*p*	*p* (Egger)	*p* (Begg)
** * Anthropometric Indicators * **									
**Overall**	13	46	−0.12 (−0.24; 0.01)	0.065	112.05	59.84	<0.0001	0.596	0.698
Excluding high-risk-of-bias studies (D1-D2 Robs tool 2)	10	43	−0.12 (−0.26; 0.02)	0.088	111.6	62.36	<0.0001	0.515	0.623
** *Target group:* **									
Only women	5	9	−0.11 (−0.34; 0.11)	0.335	6.75	0.00	0.574	0.754	0.835
Healthy	8	29	−0.12 (−0.35; 0.11)	0.301	106.89	73.81	<0.0001	0.366	0.881
Generation X (born after 1965)	8	26	−0.07 (−0.19; 0.05)	0.268	17.54	0.00	0.861	0.743	0.440
** *Intervention type:* **									
E-Health	8	22	−0.16 (−0.41; 0.1)	0.230	92.59	77.32	<0.0001	0.434	0.756
M-Health	5	24	−0.06 (−0.16; 0.03)	0.202	15.11	0.00	0.891	0.861	0.960
**Specific Outcomes**									
**BMI**	11	15	−0.43 (−1.15; 0.30)	0.247	104.71	86.63	<0.0001	0.272	0.400
**WEIGHT**	10	15	0.42 (−0.45; 1.29)	0.341	11.20	0.00	0.670	0.633	0.805
**WAIST**	5	8	−1.77 (−3.10; −0.44)	0.009	18.1	61.13	0.012	0.932	1.000
**BODY FAT %**	5	8	−0.79 (−1.63; 0.06)	0.068	16.94	58.67	0.018	0.128	0.805
			**Cohen’s d**		**Test of heterogeneity**			**Publication bias**	
** * Lifestyle * ** ** * Behaviors * **									
**Overall**	9	24	−0.20 (−0.40; 0.00)	0.052	211.45	89.12	<0.0001	0.252	0.620
Excluding high-risk-of-bias studies (D1-D2 Robs tool 2)	6	21	−0.01 (−0.16; 0.15)	0.926	93.34	78.57	<0.0001	0.973	0.856
** *Target group:* **									
Only women	2	7	−0.04 (−0.21; 0.14)	0.694	38.03	84.22	<0.0001	0.711	0.652
Healthy	5	13	−0.37 (−0.96; 0.23)	0.225	126.26	90.50	<0.0001	0.716	0.714
Generation X (born after 1965)	6	17	−0.28 (−0.52; −0.04)	0.020	168.07	90.40	<0.0001	0.187	0.805
** *Intervention type:* **									
E-Health	8	22	−0.07 (−0.25; 0.10)	0.427	133.04	84.22	<0.0001	0.750	0.978
M-Health	1	2	/	/	/	/	/	/	/
**Specific Outcomes**									
**Physical activity**	8	10	−0.01 (−0.39; 0.36)	0.939	67.26	86.62	<0.0001	0.754	0.929
**Energy intake**	4	5	−0.93 (−1.88; 0.01)	0.052	62.81	93.63	<0.0001	0.375	0.142
**Eating behavior**	4	9	−0.13 (−0.40; 0.14)	0.341	65.75	87.83	<0.0001	0.508	0.404

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
