# Peer review of "E-Health and M-Health in Obesity Management: A Systematic Review and Meta-Analysis of RCTsâ€"

_nutrients, 2025, doi:10.3390/nu17132200_

Round 1
Reviewer 1 Report
Comments and Suggestions for Authors
This review is generally well-structured and provides valuable insights into the impact of digital health interventions on obesity management.
The sentence “current treatment options have not been able to overcome the problem” is vague. Consider specifying what you mean by "overcome"—e.g., long-term weight maintenance, adherence, or widespread effectiveness.
While you mention the potential of eHealth/mHealth, the rationale could be stronger. Cite existing gaps or shortcomings in traditional interventions that digital tools might address (e.g., scalability, real-time feedback, cost-efficiency).
The cutoff date of October 4, 2024, is appropriate, but ensure it’s justified (e.g., aligned with a PROSPERO registration or a protocol).
Clarify whether interventions had to be exclusively digital or if mixed interventions (digital + in-person) were included. That could affect heterogeneity.
The only significant finding is a small reduction in waist circumference (WMD = -1.77 cm). You should comment on the clinical relevance of this change. Is it meaningful? The lack of significant effects on other anthropometric or behavioral outcomes raises questions. Why might this be? Possible reasons include: Short intervention durations; Poor adherence or engagement; Variability in the type and intensity of interventions; Self-reporting bias in behavioral outcomes
The conclusion that digital health interventions produce “modest but significant benefits” may be overstated given the singular statistically significant outcome. A more balanced statement would be that limited evidence supports modest benefits for waist circumference.
The call for more personalized approaches is valid, but this point could be expanded. How might personalization be operationalized in digital health interventions? Through AI, adaptive feedback, integration with clinician support?
Author Response
Comment 1:
"The sentence 'current treatment options have not been able to overcome the problem' is vague. Consider specifying what you mean by 'overcome'---e.g., long-term weight maintenance, adherence, or widespread effectiveness."
Response: We agree that the term "overcome the problem" was too vague and required greater scientific specificity. We have revised the sentence in the abstract to provide clear, measurable outcomes.
Comment 2:
"While you mention the potential of eHealth/mHealth, the rationale could be stronger. Cite existing gaps or shortcomings in traditional interventions that digital tools might address (e.g., scalability, real-time feedback, cost-efficiency)."
Response: We agree that our rationale needed strengthening with specific gaps that digital interventions address. We have added a comprehensive paragraph in the introduction.
Comment 3:
"The cutoff date of October 4, 2024, is appropriate, but ensure it's justified (e.g., aligned with a PROSPERO registration or a protocol)."
Response: We appreciate this methodological clarification request. The cutoff date is appropriately justified and aligned with our PROSPERO registration. We have added explicit justification in section 2.2:
Comment 4:
"Clarify whether interventions had to be exclusively digital or if mixed interventions (digital + in-person) were included. That could affect heterogeneity."
Response: This is a methodologically crucial point that affects interpretation of our heterogeneity findings. Interventions included were not exclusively digital. We have clarified inclusion criteria and acknowledged the impact on heterogeneity:
Comment 5:
"The only significant finding is a small reduction in waist circumference (WMD = -1.77 cm). You should comment on the clinical relevance of this change. Is it meaningful? The lack of significant effects on other anthropometric or behavioral outcomes raises questions. Why might this be? Possible reasons include: Short intervention durations; Poor adherence or engagement; Variability in the type and intensity of interventions; Self-reporting bias in behavioral outcomes."
Response: We thank the reviewer for this insightful comment requiring deeper analysis of clinical relevance and mechanisms underlying null effects. We have added comprehensive analysis in the discussion.
Comment 6:
"The conclusion that digital health interventions produce 'modest but significant benefits' may be overstated given the singular statistically significant outcome. A more balanced statement would be that limited evidence supports modest benefits for waist circumference."
Response: We acknowledge that our original conclusions were overstated given the limited scope of significant findings. We have revised conclusions.
Comment 7:
"The call for more personalized approaches is valid, but this point could be expanded. How might personalization be operationalized in digital health interventions? Through AI, adaptive feedback, integration with clinician support?"
Response: We agree that personalization recommendations needed concrete operationalization strategies. We have added a comprehensive section in discussion.

Reviewer 2 Report
Comments and Suggestions for Authors
Protocol registered (PROSPERO) and PRISMA-guided..important first step done right. Heterogeneity is sky-high (I² > 80% in most outcomes)..raises concerns about pooling validity. Risk of bias across studies is substantial: “High” or “Some Concerns” dominate. Key threats include unblinded outcome assessments, missing data, and variable quality of interventions. Standardization lacking: Behavior metrics (e.g., physical activity, eating behavior) varied wildly in method and units—compromises interpretability.
Waist circumference reduction is the standout: clinically relevant as a cardiometabolic risk marker. Null or borderline effects on BMI, weight, body fat, activity, and intake…not compelling for causal inference. Interventions spanned too diverse populations (from postpartum women to patients with binge eating disorder), diluting signal with noise.
Strong justification for integrating digital health into broader obesity strategies. However, effects are modest and inconsistent—currently insufficient for standalone clinical recommendation. Lack of subgroup analyses (e.g., age, BMI class, digital literacy) limits practical insights for population targeting.
What to improve:
Stratify by intervention type and target group: current pooling overheterogenizes real effects.
Consider GRADEing the evidence to enhance interpretability for clinicians and policymakers.
Provide sensitivity analyses excluding high-risk-of-bias studies.
Emphasize waist circumference more strongly—clearly the most promising outcome.
Reframe conclusions: digital tools may complement but not replace established interventions.
Discuss how intervention fidelity and digital literacy affect outcomes—currently missing.
Author Response
Comment 1:
Stratify by intervention type and target group: current pooling overheterogenizes real effects.
Response:
Thank you very much for your valuable comment. As suggested, we have stratified the analysis by intervention type and target group, as reported in the updated Table 2 and described in the meta-analysis results section.
Since digital literacy was not explicitly reported in the included studies, we considered Generation X as a proxy, in line with the relevant literature (Papp-Zipernovszky O, Horváth MD, Schulz PJ, Csabai M. Generation Gaps in Digital Health Literacy and Their Impact on Health Information Seeking Behavior and Health Empowerment in Hungary. Front Public Health. 2021 May 13;9:635943. doi: 10.3389/fpubh.2021.635943. PMID: 34055714; PMCID: PMC8158579), and this point has been addressed in the Discussion section.
Comment 2:
Provide sensitivity analyses excluding high-risk-of-bias studies
Response:
We sincerely thank you for your helpful suggestion. As requested, we conducted sensitivity analyses by excluding individual studies for Anthropometric Indicators and Lifestyle
Behaviours, as detailed in the Supplementary Material. Additionally, we performed a sensitivity analysis excluding studies at high risk of bias, specifically those rated at high risk in domains 1 and 2 of the RoB 2 tool (see Supplementary Material). These results are also reported in Table 2.
Comment 3:
Consider GRADEing the evidence to enhance interpretability for clinicians and policymakers
Response:
We appreciate this suggestion to enhance the clinical interpretability of our findings through GRADE assessment. We have conducted a comprehensive GRADE evaluation for all seven outcomes and added this assessment to our results section and to discussion.
Comment 4:
"Emphasize waist circumference more strongly—clearly the most promising outcome.
Response: we have enhanced emphasis on this result in discussion.
Comment 5:
Reframe conclusions: digital tools may complement but not replace established interventions.
Response: Thank you for this comment. We have modified conclusions in the abstract.
Comment 6:
Discuss how intervention fidelity and digital literacy affect outcomes
Response: Added comprehensive paragraph in discussion.
We believe these revisions have significantly strengthened the manuscript's scientific rigor, clinical relevance, and methodological transparency. We thank both reviewers for their valuable contributions to improving this work.

Round 2
Reviewer 1 Report
Comments and Suggestions for Authors
After the major modifications to the manuscript, I believe it's publishable.
Reviewer 2 Report
Comments and Suggestions for Authors
All my comments were addressed appropriately. Revisions were incorporated into the manuscript in a transparent and methodologically sound manner, thereby significantly improving its quality. I have no further comments.